# Quantifying Data Dependencies with Rényi Mutual Information and Minimum Spanning Trees

**DOI:** 10.3390/e21020100

**Published:** 2019-01-22

**Authors:** Anne Eggels, Daan Crommelin

**Affiliations:** 1Centrum voor Wiskunde & Informatica, 1090 GB Amsterdam, The Netherlands; 2Korteweg-de Vries Institute for Mathematics, University of Amsterdam, 1090 GE Amsterdam, The Netherlands

**Keywords:** Rényi entropy, dependent data, large datasets, minimum spanning trees

## Abstract

In this study, we present a novel method for quantifying dependencies in multivariate datasets, based on estimating the Rényi mutual information by minimum spanning trees (MSTs). The extent to which random variables are dependent is an important question, e.g., for uncertainty quantification and sensitivity analysis. The latter is closely related to the question how strongly dependent the output of, e.g., a computer simulation, is on the individual random input variables. To estimate the Rényi mutual information from data, we use a method due to Hero et al. that relies on computing minimum spanning trees (MSTs) of the data and uses the length of the MST in an estimator for the entropy. To reduce the computational cost of constructing the exact MST for large datasets, we explore methods to compute approximations to the exact MST, and find the multilevel approach introduced recently by Zhong et al. (2015) to be the most accurate. Because the MST computation does not require knowledge (or estimation) of the distributions, our methodology is well-suited for situations where only data are available. Furthermore, we show that, in the case where only the ranking of several dependencies is required rather than their exact value, it is not necessary to compute the Rényi divergence, but only an estimator derived from it. The main contributions of this paper are the introduction of this quantifier of dependency, as well as the novel combination of using approximate methods for MSTs with estimating the Rényi mutual information via MSTs. We applied our proposed method to an artificial test case based on the Ishigami function, as well as to a real-world test case involving an El Nino dataset.

## 1. Introduction

In the field of uncertainty quantification (UQ), the question to what extent random variables are dependent emerges in various places. In UQ, the uncertainties in simulation models of complex systems are explored, for example by assessing what the probability distribution of the simulation output is given the probability distribution(s) of various uncertain model input variables [1,2,3]. Many methods in UQ are designed for situations where the random input variables are mutually independent, hence the (in)dependence of these inputs is of obvious importance.

A topic in UQ that is particularly relevant for this study is sensitivity analysis, where it is investigated which random inputs induce the largest uncertainties in the simulation output [4,5,6]. Sensitivity analysis is closely related to the question how strongly dependent the output is on the individual input variables. By quantifying these dependencies, one can order the inputs by the extent to which the output is dependent on them (from strongly to weakly dependent), providing relevant information for sensitivity analysis.

In this study, we consider quantification of dependencies between random variables in situations where: (i) the distribution of the random variables is unknown and only a given, finite sample from the joint distribution is available; and (ii) the dependencies can potentially be non-linear, implying that the underlying joint distribution is non-Gaussian. It is well-known that information theory forms a suitable framework to deal with non-Gaussian distributions and nonlinear dependencies [7]. However, information-theoretic quantities such as Shannon entropy and Kullback–Leibler divergence are formulated in terms of probability distributions or probability density functions, precluding their exact computation if the distribution is unknown.

As an alternative, one can estimate entropy (and related quantities) from sample data. Such estimation is nontrivial however. A typical approach is to use a plug-in estimator for the probability density (e.g., by binning or kernel density estimation) to compute (an estimate of) the entropy. This approach faces problems relating to the difficulties of estimating densities (see, e.g., [8] for a discussion). To bypass these problems, Hero et al. [8,9,10] proposed an appealing and elegant method for estimating entropies from data by computing minimum spanning trees (MSTs) of the data and using the length of the MST in an estimator for Rényi entropy. This method does not require estimation of probability densities. In this study, we employ the method from [8,9,10] for quantifying dependencies, by using the length of the MST as a measure of dependence between random variables.

In principle, MSTs can be computed exactly, with algorithms due to Kruskal [11] and Prim [12]. However, these algorithms become expensive in the case of large datasets. Therefore, we test several approximate methods to compute the MST length with strongly reduced computational cost.

Our main contribution is to accelerate the method in [9] by computing an approximate rather than an exact MST. In the era of data science, it is expected large datasets will appear where a speedup of the computations is desirable. The question herein is whether the approximation is accurate enough to replace the costly but exact MST. Our second contribution is that we show it is not necessary to compute the Rényi divergence to quantify dependence for the purpose of ranking dependency strength between different random variables. An estimator derived from it is equally informative yet easier to compute.

Although the present study is concerned primarily with dependency analysis rather than sensitivity analysis, the latter is an important motivation for the work reported here. We will employ the tools developed here for sensitivity analysis in a future study, and note that the relevance of entropy estimation for sensitivity analysis has been discussed before (e.g., [13,14]). However, these studies rely on density estimation to compute entropy. The MST approach to entropy estimation from [8,9,10] has not been considered before in the context of dependency analysis and sensitivity analysis.

This paper is organized as follows: In Section 2, we summarize some theory regarding entropy and we relate Rényi entropy to Rényi divergence. Furthermore, we discuss the estimation of Rényi entropy with the MST. Next, in Section 3, the approximation methods for computing the MST are discussed. Section 4 consists of validation of the proposed estimator and determining the consistency and robustness of the approximations to it. In Section 5, we apply the proposed methods to several test cases, including one with data from atmosphere–ocean observations. Section 6 concludes.

## 2. Dependence and Entropy

We summarize a few basic definitions and concepts in Section 2.1 and Section 2.2. Section 2.3 describes the estimator of Rényi entropy as developed in [8]. This paves the way for defining our proposed estimator for dependency between random variables in Section 2.4. This estimator is related to, but not equal to the Rényi mutual information. Section 2.5 shows a proof-of-concept of this estimator.

### 2.1. Rényi Entropy and Divergence

Rényi entropy is a generalization of the (continuous version of) the Shannon entropy. The latter, also known as differential entropy, is defined as (e.g., [7])
(1)$$H\left(X\right)=-\int_{\Omega }p\left(x\right)\log\left(p\left(x\right)\right)dx,$$
where p(x) is the probability density function of the continuous random variable *X*, and Ω is the domain of *X*. The generalization by Rényi reads
(2)$$H_{\alpha }\left(X\right)=\frac{1}{1-\alpha }\log\left(\int_{\Omega }{\left(p(x)\right)}^{\alpha }dx\right),$$
for α∈(0,∞) (see [7]). In the limit α→1, the Rényi entropy in Equation (Equation 2), sometimes referred to as α-entropy, converges to the differential entropy in Equation (Equation 1).

Related to the Rényi entropy in Equation (Equation 2) is the Rényi divergence [15] of a probability density function f(x) with respect to another probability density function g(x):
(3)$$D_{\alpha }\left(f,g\right)=\frac{1}{\alpha -1}\log\left(\int_{\Omega }{\left(\frac{f(x)}{g(x)}\right)}^{\alpha }g\left(x\right)dx\right).$$


This divergence is well-defined if *g* dominates *f* (i.e., $f(x^{*})=0$ for all $x^{*}$ where $g(x^{*})=0$). It converges to the Kullback–Leibler (KL) divergence $D_{\mathrm{KL}}\left(f,g\right)=\int f\left(x\right)\log\left(f\left(x\right)/g\left(x\right)\right)dx$ if α→1. If f=g almost everywhere, we have $D_{\alpha }\left(f,g\right)=0$, otherwise $D_{\alpha }\left(f,g\right)>0$.

### 2.2. Entropy as Measure of Dependence

It is well-known that the dependence between two random variables *Y* and *Z* can be characterized by their mutual information, i.e., the difference between the Shannon entropy of (Y,Z) jointly and the sum of the Shannon entropies of *Y* and *Z* individually,
(4)I(Y,Z)=H(Y)+H(Z)−H(Y,Z),
where H(Y,Z) is the joint entropy of *Y* and *Z* (see [7]). Shannon entropy can also be defined by the KL divergence of the joint density g(y,z) with respect to $\tilde{g}\left(y,z\right)$, the product of the marginal densities $g_{y}\left(y\right)$ and $g_{z}\left(z\right)$ (where $g_{y}\left(y\right)=\int g\left(y,z\right)dz$ and $g_{z}\left(z\right)=\int g\left(y,z\right)dy$).

We can generalize this information-theoretic characterization of dependence by using Rényi rather than KL divergence. Thus, we quantify dependence by $D_{\alpha }\left(g,\;\tilde{g}\right)$, the Rényi divergence of g(y,z) with respect to $\tilde{g}\left(y,z\right)$. This can also be used to define the Rényi mutual information [16,17] as:
(5)$$I_{\alpha }{\left(Y,Z\right)}_{=}\frac{1}{\alpha -1}\log\left(\int_{Y}\int_{Z}{\left(\frac{p_{y,z}\left(y,z\right)}{p_{y}\left(y\right)p_{z}\left(z\right)}\right)}^{\alpha }p_{y}\left(y\right)p_{z}\left(z\right)dzdy\right)$$


We note that in the limit α→1, $I_{\alpha }\to I$. It is easy to verify that, by construction, $p_{y}\left(y\right)p_{z}\left(z\right)$ dominates $p_{y,z}\left(y,z\right)$, hence the Rényi and KL divergences are well-defined. As was shown in [18], the divergence $D_{\alpha }\left(g,\;\tilde{g}\right)$ is equivalent to the Rényi entropy of a different density, resulting from a coordinate transformation. Let $\left(y^{\prime },z^{\prime }\right)=G\left(y,z\right)$ where $G:R^{2}\mapsto R^{2}$ is the Rosenblatt transformation [19,20] induced by the product density $\tilde{g}$ (i.e., the transformation that uniformizes $\tilde{g}$). The joint density induced by this transformation is $h(y^{\prime },z^{\prime })$, given by
(6)$$h\left(y^{\prime },z^{\prime }\right)=\frac{g(G^{-1}\left(y^{\prime },z^{\prime }\right))}{\tilde{g}\left(G^{-1}\left(y^{\prime },z^{\prime }\right)\right)}$$
and as a result,
(7)$$D_{\alpha }\left(g,\;\tilde{g}\right)=\frac{1}{\alpha -1}\log\left(\int {\left(\frac{g(y,z)}{\tilde{g}\left(y,z\right)}\right)}^{\alpha }\tilde{g}\left(y,z\right)\;dydz\right)=\frac{1}{\alpha -1}\log\left(\int h^{\alpha }\left(y^{\prime },z^{\prime }\right)\;dy^{\prime }dz^{\prime }\right)=-H_{\alpha }\left(h\right).$$


Thus, $D_{\alpha }\left(g,\;\tilde{g}\right)$ equals $-H_{\alpha }\left(h\right)$, the Rényi entropy of the induced joint density *h* given in Equation (Equation 6), up to a minus sign. This makes sense because a stronger dependence is coherent with a smaller $H_{\alpha }\left(h\right)$ and a larger $D_{\alpha }\left(g,\tilde{g}\right)$. In addition, we note that Equation (Equation 6) is similar to the copula density (see also [16]).

If the density g(y,z) (and hence also $\tilde{g}\left(y,z\right)$) is unknown and only a sample of its probability distribution is available, we can (approximately) carry out the coordinate transformation *G* induced by $\tilde{g}$ without having to estimate *g* or $\tilde{g}$ itself. Instead, we apply the rank transform [21,22], together with centering, to the data. The marginals of the rank-transformed data are discrete representations of the uniform distribution U[0,1].

In summary, if we are given a sample of the random variables Y,Z with joint probability density g(y,z), we can estimate the Rényi divergence $D_{\alpha }\left(g,\tilde{g}\right)$ by first applying the rank-transform to the sample data, and then estimating the Rényi entropy of the transformed data $H_{\alpha }\left(h\right)$.

Before turning to the estimation of the Rényi entropy, we conclude this subsection by pointing out the connection to another measure of divergence or dissimilarity between probability densities, the Hellinger distance, and by commenting on the choice of α. The Hellinger distance [23] can be computed from $D_{1/2}$ by easy transformations. The advantages of using $D_{1/2}$ are that it is a distance metric [24] and it is optimal in the context of classification problems [8,24].

### 2.3. Estimator of the Rényi Entropy

It is not straightforward how to estimate the entropy of a given dataset. As discussed in the Introduction, one approach is to estimate the probability distribution or density underlying the dataset, and compute the entropy from this estimated density. Both parametric and non-parametric methods (e.g., kernel-density estimation and binning) can be used to estimate the distribution, however the results are quite sensitive to the details of the method used [25]. See also the work in [8] where the difficulties of entropy estimation by means of plug-in density estimators are summarized. Noshad et al. [26] circumvented the plug-in density and proposed a direct estimation method based on a graph-theoretical interpretation. Very recently, Moon et al. [17] derived mean squared error convergence rates of kernel density-based plug-in estimators of mutual information and proposed ensemble estimators that achieve the parametric rate, although there are several restrictions on the densities and the kernel used.

To circumvent the need for density estimation, we employ a different approach to estimate entropy in this study, one in which the sample data is used directly. This approach is due to Hero and Michel [8,9,10], who proposed a direct method to estimate the Rényi entropy, based on constructing minimal graphs spanning the data points. This approach can be used to estimate $H_{\alpha }$ with 0<α<1, hence it does not apply to estimation of the Shannon entropy. This motivates our focus on the Rényi entropy here.

Let $X_{N}$ denote a sample of the multivariate random variable *X* with sample size *N*. The dimension of the domain of *X* is *d*. The estimator proposed in [8,9,10] is
(8)$${\hat{H}}_{\alpha }\left(X_{N}\right)=\frac{1}{1-\alpha }\left(\log\left(\frac{L_{\gamma }\left(X_{N}\right)}{N^{\alpha }}\right)-\log\beta_{L,\gamma }\right)=\frac{1}{1-\alpha }\log\left(\frac{L_{\gamma }\left(X_{N}\right)}{\beta_{L,\gamma }\;N^{\alpha }}\right),$$
where α=(d−γ)/d and $\beta_{L,\gamma }$ is a constant only depending on γ and the definition of $L_{\gamma }$. The functional $L_{\gamma }\left(X_{N}\right)$ is defined as
(9)$$L_{\gamma }\left(X_{N}\right)=\min_{T(X_{N})}\sum_{e\in T(X_{N})}{\left|e\right|}^{\gamma },$$
where $T(X_{N})$ is the set of spanning trees on $X_{N}$ and *e* denotes an edge. The parameter γ can be freely chosen within the interval (0,d), with d≥2. The choice of γ determines α, and α∈(0,1) if γ∈(0,d). The estimator in Equation (Equation 8) is asymptotically unbiased and in fact strongly consistent for α∈(0,1) [18]. If γ is chosen to be 1 and |e| denotes the Euclidean distance between data points in $X_{N}$, then $L_{\gamma }\left(X_{N}\right)$ describes the length of the minimum spanning tree (MST) on the dataset. In this study, we focus on d=2, but we note that Equation (Equation 8) applies to situations with d>2 as well. With d=2 and γ=1, we have α=1/2, a value for α whose theoretical motivation is discussed above.

We give an illustration of the MST on data in the unit square, i.e., d=2 (we recall that data sampled from distributions on other domains will be mapped to the unit (hyper)square by the rank transform). In the case of independence, the MST will approximately cover the full domain. In the case of dependence, the density is manifestly nonuniform. As a result, the average edge length decreases and so does the total length of the MST. This is illustrated in Figure 1. A shorter MST length (with *N* constant) implies lower entropy (cf. Equation (Equation 8)), and therefore stronger dependence. Note that there is no assumption on the shape or structure of the dependence.

### 2.4. Quantifier of Dependence

The constant $\beta_{L,\gamma }$ in Equation (Equation 8) is unknown. However, it does not depend on the distribution of *X* and can be regarded as a constant offset or bias correction. When comparing the entropies of two random variables $X^{\left(1\right)}$ and $X^{\left(2\right)}$ with $\dim\left(X^{\left(1\right)}\right)=\dim\left(X^{\left(2\right)}\right)=d$, the difference ${\hat{H}}_{\alpha }\left(X_{N}^{\left(1\right)}\right)-{\hat{H}}_{\alpha }\left(X_{N}^{\left(2\right)}\right)$ does not depend on $\beta_{L,\gamma }$ provided the same type of spanning tree is used (i.e., same functional *L* and γ).

This brings us to one of the contributions of this study. We propose using the following quantity to measure (or quantify) dependence:
(10)$$H_{\alpha }^{*}\left(Z_{N}\right)=\log\left(\frac{L_{\gamma }\left(Z_{N}\right)}{N^{\alpha }}\right)$$
where $Z_{N}$ is a dataset consisting of the rank-transform of *X*, containing *N* data points. $H_{\alpha }^{*}\left(Z_{n}\right)$ equals $\left(1-\alpha \right){\hat{H}}_{\alpha }\left(Z_{N}\right)$ (Equation (Equation 8)) when $\beta_{L,\gamma }=1$. We set α=1/2, as discussed in the previous section. We focus here on the case d=2 so that γ=1. The advantage of this estimator is that, in the case where multiple dependencies are compared, the value of $\beta_{L,\gamma }$ is not necessary to obtain an ordering of them, and, therefore, this value does not need to be estimated.

Besides obtaining an ordering in terms of the dependency strength, we can use Equation (Equation 10) to construct a reference level for distinguishing between independent and dependent variables. This construction consists of computing $H_{\alpha }^{*}\left(Z_{N}\right)$ on *r* different datasets of rank-transformed bivariate uniformly distributed data. These datasets are generated independently of each other, and, within each dataset, the two variables are independent as well (as they have a bivariate uniform distribution). This is done by sampling 2N values from the one-dimensional uniform distribution and reshaping them into a N×2 dataset. From this, we obtain an empirical distribution, based on *r* samples, for $H_{\alpha }^{*}\left(Z_{N}\right)$ for independent variables (we recall that data from independent variables always leads to data uniformly distributed on a grid after the rank-transform). The reference level can then be defined as the η-quantile of the empirical distribution for η small, since $H_{\alpha }^{*}\left(Z_{N}\right)$ decreases with increasing dependence. This leads to the following statistical test for dependence. The null hypothesis is that the random variables are independent, which implies this null hypothesis is rejected if $H_{\alpha }^{*}\left(Z_{N}\right)$ is significantly smaller (more negative) than could be expected if the variables were independent. Thus, the null hypothesis is rejected if
(11)$$H_{\alpha }^{*}\left(Z_{N}\right)\leq \eta ,$$
in which η is the 0.01 or 0.05 quantile of the empirical distribution for $H_{\alpha }^{*}\left(Z_{N}\right)$ for independent variables. In the remainder of this text, we write $H_{\alpha }^{*}\left(Z\right)$ instead of $H_{\alpha }^{*}\left(Z_{N}\right)$, as we continue with data only.

### 2.5. Proof of Concept

First, we computed the quantifier of dependence (Equation (Equation 10)) on multiple datasets with varying distributions to analyze its behavior. Then, its convergence and robustness for increasing values of *N* were studied. For the distributions considered in this section, it was possible to compute the Rényi entropy with high accuracy for the scaled (i.e., rank-transformed) distribution such that a comparison can be made between the behaviors of Equations (Equation 10) and (Equation 2).

We used datasets sampled from the following distributions: (i) a bivariate uniform distribution; (ii) a standard normal distribution with varying correlation coefficient ρ; (iii) a constant density on a region within the unit square with area 1−A, in which the data are focused in the lower left corner (corner distribution); and (iv) a constant density on a region within the unit square with area 1−A, but in which the data are focused on a symmetry axis (line distribution). For the latter two cases, the region includes values near the minimum and maximum of both variables, such that the ranges of the region for varying *A* stay the same. These distributions are also referred to as shape distributions. For each of the four distributions, $r=10^{2}$ datasets were generated with $N=10^{3}$ data points each. ρ and *A* were varied in steps of 0.10 from 0.05 to 0.95. We would like to emphasize this number of datasets can be different from the *r* one may use to obtain a reference level to distinguish between independent and dependent variables. Here, *r* is the number of datasets used to show the effect of sample variation.

In Figure 2, the empirical cumulative distribution function (CDF) of $H_{\alpha }^{*}\left(Z\right)$ (Equation (Equation 10)) is plotted for data from the bivariate uniform distribution. It can be seen that the distribution is concentrated in a rather narrow interval. Finally, we explored how this interval becomes more or less narrow as the number of data (N) changes, of which the results are given at the end of this Section.

In Figure 3, the empirical CDF of $H_{\alpha }^{*}\left(Z\right)$ is plotted for both the normal distribution (Case (ii)) and the two shape distributions (Cases (iii) and (iv)), for different small values of ρ and *A*. More precisely, for each dataset sampled from one of these distributions, we first applied the rank-transform and then evaluated Equation (Equation 10) for the rank-transformed data. The empirical CDF for the uniform distribution, as shown in Figure 2, is included in black for comparison. If ρ=0 or A=0, there is no longer a dependence and the empirical CDF obtained from the normal distribution or shape distributions coincides with the empirical CDF from the bivariate uniform distribution (modulo differences due to finite sample size, N<∞).

It can be seen that, for the shape distributions, the quantifier of dependence is more distinctive for small values of *A* than it is for the normal distribution with small ρ. Hence, weak dependencies in the shape distributions can more easily be detected than in the normal distribution. To demonstrate the behavior of $H_{\alpha }^{*}\left(Z\right)$ for the full range of values of ρ and *A*, we plot the mean together with the empirical 95% confidence intervals in Figure 4b. In the case of ρ=A=0, the distribution of the uniform (independent) distribution is recovered, indicated by a different color.

In Figure 4a, it can be seen why the differences between the CDFs are small in the case of the normal distribution: for the normal distribution, the entropy is nearly flat as a function of ρ, for small ρ values. Figure 4b shows the rank-transform has a more severe effect on the corner than on the line distribution, since the estimates for the corner distribution decrease more slowly. The distortion of the shape of the graph for the corner distribution between Figure 4a,b was caused by the distortion of the distances between data points after the rank-transform. Note that both shape distributions have the same Rényi entropy if the rank-transform is not applied. The effect of the rank-transform in this case can be seen in Figure 5. The closer *A* is to 1, the more the data fall in the two boxes for the corner distribution, due to the combination of skewness and discontinuity. This does not hold for the line distribution. In the limit of A→1 and N→∞, the entropy of the rank-transformed corner distribution goes to −log(2), while it goes to −∞ for the rank-transformed line distribution. Hence, it is consistent that $H_{\alpha }^{*}\left(Z\right)$ does not go to −∞ for the corner distribution. We note that the shape distributions are quite artificial and have discontinuous density, while datasets in practice are usually samples from a distribution with a smooth density and a less artificial shape.

We conclude this section by investigating the effect of varying the size of the dataset, *N*. We computed the empirical CDF of $H_{\alpha }^{*}\left(Z\right)$ using N=10, $10^{2}$ and $10^{3}$. For this computation, we increased *r* to $10^{4}$ to obtain a smooth CDF. The resulting empirical CDFs are shown in Figure 6. It can be seen that the distribution becomes narrower with increasing *N*, as expected. Thus, larger *N* makes the comparison of estimates easier due to the smaller confidence intervals involved.

We note that, for higher values of *N*, the computations become expensive due to the computation of the edge lengths before computing the MST. The order of this operation is $O(N^{2})$. Kruskal’s method for computing MSTs [11] runs in $O(N^{2}\log\left(N\right))$ time, but the steps are faster than the ones needed for computing the edge lengths. Prim’s method for computing MSTs [12] can be faster ($O(N^{2}+N\log\left(N\right))$), but the implementation is more involved. The high cost of computing the exact MST motivates us to investigate approximate algorithms in the next section.

## 3. Approximation Methods

For large datasets (i.e., large *N*), the computational cost of evaluating the estimator in Equation (Equation 10) becomes prohibitively high, due to the computational complexity of constructing the MST. In this section, we discuss methods to reduce the computational cost by approximating the value of Equation (Equation 10) evaluated on a large dataset. We consider three types of approximation: First, MSTs can be computed on multiple subsets of the data. The second type aggregates data points into clusters, such that only one MST calculation is performed on the cluster centers. The third type clusters the data points, constructs MSTs on each cluster and combines them in a smart way [27].

### 3.1. Sampling-Based MST

In this method, the dataset is split into *K* subsets and $H_{\alpha }^{*}\left(Z\right)$ (Equation (Equation 10)) is computed on each of the subsets. Thus, *K* estimates of $H_{\alpha }^{*}\left(Z\right)$ are obtained. Their arithmetic mean becomes the new estimate, while their variance is a measure for the quality of the new estimate. The number *K* can be chosen by the user, although it represents a trade-off between accuracy and computation time (since both accuracy and computational time decrease with increasing *K*).

The splitting can be done in various ways, of which random and stratified are the most straightforward ones. In the random splitting, data points are allocated randomly to one of the *K* subsets, which all have size N/K (rounding is neglected). In the proportional splitting, *k* clusters are generated with the *K*-means method [28,29] and these are considered the strata. The data points in each stratum are then proportionally allocated to the *K* subsets. The number of clusters involved (*k*) would be another parameter, which we chose equal to N/K.

### 3.2. Cluster-Based MST

Another way of reducing the computational burden would be to cluster the data into many clusters and compute the MST on the cluster centers. In this case, the clustering method can be chosen, as well as the number of clusters. To be consistent with the previous method, we defined here the number of clusters to be k=N/K, such that the number of points in the MST was similar to the previous method. Furthermore, it is possible to include the size (weight) of the clusters as well. Two different clustering methods were investigated: *K*-means [29] and PCA-based clustering [30]. The construction of the MST was performed both without and with weighting. The weighting is harmonic, which implies that, in the construction of the MST, the edge between data points *i* and *j*, $|e_{ij}|$, is replaced by
(12)$$W_{ij}\left|e_{ij}\right|,\;W_{ij}=\frac{2k}{\frac{1}{w_{i}}+\frac{1}{w_{j}}},$$
where *k* is the number of cluster and $w_{i}$, $w_{j}$ are the weights of the clusters, computed by the fraction of data points in that cluster.

### 3.3. Multilevel MST

This method was developed by Zhong et al. [27] and is called FMST (fast MST). It is based on the idea that, to find a neighbor of a data point, it is not necessary to consider the complete dataset of size *N*. In the FMST method, the dataset is partitioned in $\sqrt{N}$ clusters via the *K*-means clustering method, and MSTs are constructed on each of the clusters. In a next step, a MST is constructed on the cluster centers to determine which clusters get connected at their boundaries. Between these clusters, the shortest connecting edges are computed. Because this is heuristic, the complete process is repeated with the midpoints of the edges connecting different MSTs being the initial cluster centers for the *K*-means method. The resulting two MSTs are merged into a new graph and its MST is computed as final outcome of the method. Zhong et al. [27] reported good results with this method, which has a computational complexity of $O(N^{1.5})$. Errors occur only if points that should be connected end up in different clusters twice, which does not occur often and has only a minor effect. For high-dimensional datasets, erroneous edges are included more often, but the effect thereof is smaller. Both are due to the curse of dimension.

### 3.4. Comparison

The accuracy, robustness and computational time of the methods explained before were tested on r=50 bivariate uniform datasets with $N=10^{4}$ and r=60 bivariate strongly dependent datasets with $N=10^{4}$. The strongly dependent datasets were chosen as projections of the dataset used in Section 5.1. Hence, in the dataset exists combinations of $x_{1}$, $x_{2}$, $x_{3}$ and $x_{5}$. Since this leads to six projections per dataset, we chose r=60 in this case. In these cases, it is computationally expensive but still feasible to compute the full (exact) MST, and we did so to be able to compare results from the three approximate methods with the method based on the full MST. For the approximate methods, we chose the parameter *K* to be 10. The results can be found in Figure 7. We mention here the effect of the implementation of *K*-means, as the choice of initialization can affect the results. We used the *K*-means++ initialization algorithm, repeated the clustering 10 times from different initializations and selected the result with the best clustering criterion.

The error estimate was computed as the absolute difference between the approximated and exact value of $H_{\alpha }^{*}\left(Z\right)$. The FMST has consistently small error, while other methods have larger error. In our opinion, it is important that the approximation is accurate and robust, i.e., has little variation in its error over multiple runs of data with the same distribution. Furthermore, it should perform well for both dependent and independent data. Therefore, we propose to use the FMST to compute approximations to the MST in the case where the dataset is large (e.g., $N\geq 10^{4}$). This is a main contribution of this paper. We have now shown that, in the computation of the Rényi mutual information, which can be done by the method in [8], it is possible to replace the MST by the FMST.

## 4. Validation of the Proposed FMST Estimator

We further investigated using the FMST method with the estimator in Equation (Equation 10). First, the effect of the size of the dataset *N* was investigated together with the robustness of the FMST estimator for relatively small *N*. Then, the FMST estimator for dependence was tested for behavior and consistency using datasets sampled from the three distributions considered before (uniform, normal and shape distributions). Furthermore, we investigated the behavior of the dependence measure obtained with MST and FMST to the exact value in a separate simulation study in Section 4.1.

It is straightforward that approximations to the MST using the correct edge distances overestimate the length of the MST. Therefore, we compared the empirical distributions of $H_{\alpha }^{*}\left(Z\right)$ based on the MST (Figure 6) to the ones based on the FMST in Figure 8. The number of repetitions is $r=10^{4}$ for $N=10^{2}$ and $10^{3}$ and $r=10^{2}$ for $N=10^{4}$ and $N=10^{5}$. It can be seen that the distributions are different, but only shifted. Furthermore, the width of the distribution decreases to almost zero for $N=10^{4}$ and $N=10^{5}$. This means the estimator is robust to sampling effects. Note that there is still a difference between the distributions for $N=10^{4}$ and $N=10^{5}$. This is due to the bias caused by approximating the MST, which reduces for increasing *N*.

We repeated the experiment in Section 2.5, but with the estimator based on FMST to evaluate the effect of the FMST approximation on different data distributions. The results are shown in Figure 9. Again, the behavior of the estimator based on FMST was the same as for MST, but only shifted upwards. This bias is very small for the line distribution. In these tests, $N=10^{3}$ to compare them to the previous results.

Based on these results, we conclude that the FMST estimator is a good and robust approximation of the MST estimator. It has a positive bias, but, for the purpose of comparing and ranking strength of dependencies as quantified by the FMST estimator, this bias does not pose a problem.

### 4.1. Comparison with the Exact Value of the Rényi Divergence

In this section, we present a simulation study to compare the true value of the dependence measure with the measures obtained with MST and FMST estimators. A direct method to estimate the Rényi mutual information based on a kernel density estimator was given by Moon et al. [17], which was used for comparison.

The distribution under consideration is
$$X=\left(\begin{array}{c} X_{1}\\ X_{2} \end{array}\right),\;X∼N\left(\mu ,\Sigma \right),\;\mu =\left(\begin{array}{c} 0\\ 0 \end{array}\right),\;\Sigma =\left(\begin{array}{cc} 1 & \rho \\ \rho & 1 \end{array}\right),$$
for which the Rényi mutual information is given by
(13)$$I_{\alpha }=\frac{-1}{2(1-\alpha )}\left(\alpha \log\left(1-\rho^{2}\right)-\log\left(1-\alpha^{2}\rho^{2}\right)\right).$$


We varied the value of ρ and studied the behavior of the MST and FMST estimator of the Rényi mutual information. A direct estimate was obtained via the estimator of Moon et al. [17] with a Gaussian kernel and a bandwidth computed by Scott’s rule [31].

Multiple sample sizes were included and the estimation was repeated *r* times, each time with newly generated data. For $N\in \{10,10^{2},10^{3}\}$, $r=10^{2}$, while r=10 for $N=10^{4}$. In the case of $N=10^{4}$, the regular MST was too expensive to perform often. We give the average value in Figure 10 in solid markers. The open markers represent the empirical 95% confidence interval for $N\in \{10,10^{2},10^{3}\}$ and the minimum and maximum values for $N=10^{4}$. The constant $\beta_{L,\gamma }$ was computed for the same *r*.

It can be seen KDE underestimates the divergence for small *N* in general. For $N=10^{4}$, ρ = −0.5, the mean for KDE seems to be missing. This is because its value was influenced by one of the estimates being −21.16. For high *N*, FMST performed better than KDE. In Figure 11, the time aspects of the different methods are considered. MST is left out for $N=10^{4}$ because of its high computational cost.

For larger values of *N*, the time complexities of FMST and KDE are as expected, with complexities $O(N^{1.5})$ and $O(N^{2})$, respectively. FMST becomes cheapest for $N=10^{4}$ while its results are very similar. Therefore, it is beneficial to use FMST in the case of large datasets.

## 5. Test Cases

The proposed methods were applied to two test cases. In the first case, we used the Ishigami function [32] and evaluated it using randomly sampled synthetic data as inputs. We considered data from two different input distributions, one without dependencies (uniform) and one with strong dependencies. On all combinations of variables (input and output), the dependence was quantified and, for the uniform dataset, compared to the values of the Sobol (total) indices. In the second test case, we investigated the El Nino dataset obtained from the UCI Machine Learning Repository.

### 5.1. Ishigami Function

This test function is from the work of Ishigami and Homma [32] and its Sobol (total) indices [33] can be computed analytically. The test function is given by
(14)$$x_{5}\left(x_{1},x_{2},x_{3}|a,b\right)=\left(a+bX_{3}^{4}\right)\sin\left(X_{1}\right)+a\sin^{2}\left(X_{2}\right),$$
where
(15)$$X_{i}=-\pi +2\pi x_{i}.$$


Parameters *a* and *b* were chosen to be 7 and 0.1, respectively, in accordance with [34]. One dataset is four-dimensional and uniformly distributed, while the other one is generated as
(16)$$x∼U\left(-2,2\right),\;x_{2}∼x_{1}^{2}+\frac{1}{2}N\left(0,1\right),\;x_{3}∼x_{1}^{3}+\frac{1}{2}N\left(0,1\right),\;x_{4}∼x_{1}^{4}+\frac{1}{2}N\left(0,1\right),$$
and range-normalized to the unit hypercube. Note that one variable (*u*) was not used in the Ishigami function. This is on purpose to show the effect of confounders. The MSTs were approximated with both the MST and the FMST method and Equation (Equation 10) was used to compute the dependencies both between input variables and between the input variables and the output variable. The ordering was $\left(x_{1},x_{2}\right)$, $\left(x_{1},x_{3}\right)$, $\left(x_{1},x_{4}\right)$, $\left(x_{1},x_{5}\right)$, etc. Because of this ordering, the pairs numbered 4, 7, 9 and 10 represented combinations of one input and the output variables ($\left(x_{1},x_{5}\right),\left(x_{2},x_{5}\right),\left(x_{3},x_{5}\right),\left(x_{4},x_{5}\right)$, respectively).

The datasets were generated r=10 times with $N=10^{4}$ samples each and the results are in Figure 12. First, it can be seen that the estimates are robust, as for each set (or pair) the estimates are all in a narrow interval (indicated by the minimum and maximum estimates).

Furthermore, it can be seen that, for the uniform dataset (Figure 12a), only the input variables $x_{1}$ and $x_{2}$ were found to have an effect on the output. For most sets, the dependence estimate is slightly below −0.4, the value attained in the case of independence. One can compare this with Figure 8, where it can be seen that, in the case of two uniformly distributed independent variables, the estimate is slightly below −0.4 in the case of $N=10^{4}$. Indeed, the input variables are all independent in the case of this uniform dataset. In addition, by construction, the Ishigami function does not depend on $x_{4}$. The independence of $x_{3}$ and $x_{5}$ is nontrivial, however it is consistent with the analysis using Sobol indices (as discussed below). $x_{5}$ is not directly dependent on $x_{3}$, although there is an interaction effect with $x_{1}$. This interaction effect is not detected with the analysis of pairwise dependencies.

The exceptions to independence are sets 4 and 7 (pairs $\left(x_{1},x_{5}\right)$ and $\left(x_{2},x_{5}\right)$), for which the $H_{\alpha }^{*}\left(Z\right)$ estimate is lower, indicating dependence. The dependence of $x_{2}$ and $x_{5}$ is stronger than between $x_{1}$ and $x_{5}$, visible in the smaller value for set 7. The (in)dependence of *I* on the various inputs is illustrated in Figure 13 where scatter plots of the input–output pairs are shown.

For the strongly dependent dataset, the analysis is less straightforward. Again, the combinations $\left(x_{1},x_{5}\right)$ and $\left(x_{2},x_{5}\right)$ are strongly dependent, while $\left(x_{3},x_{5}\right)$ shows a weak dependence. Because of the structure of the dataset, all input variables are mutually dependent. Furthermore, the dependencies within the input data are stronger than the dependencies of input variables with the output, except for the $\left(x_{2},x_{5}\right)$ combination. The relative ordering of dependencies (strongest for $\left(x_{2},x_{5}\right)$, weaker for $\left(x_{1},x_{5}\right)$, and very weak for $\left(x_{3},x_{5}\right)$ and $\left(x_{4},x_{5}\right)$) is consistent with the definition of the Ishigami function and the intuition gleaned from the right panels of Figure 13.

We continue by comparing the means of the estimates for the input–output combinations for the uniform dataset based on FMST with the values of its Sobol indices and Sobol total indices in Table 1.

For the Ishigami function, the sum of all $S_{i}$ and first-order interaction terms $S_{ij}$ equals 1, and only $S_{1}$, $S_{2}$ and $S_{13}$ are nonzero. Since $S_{2}=S_{T2}$, $x_{2}$ is not involved in interaction terms, while $S_{3}$ does not have an effect by itself. The term $S_{13}$ contributes to both $S_{T1}$ and $S_{T3}$. From the numerical values for the Sobol indices, summarized in Table 1, we conclude that $x_{2}$ is most important by itself, while $x_{1}$ is most important when interactions with other variables are included.

The proposed estimator $H_{\alpha }^{*}\left(Z\right)$ behaves similarly to $S_{i}$. It is approximately −0.42 in the case of independence (exact value depends on *N*, higher *N* leads to a smaller value, see Figure 8), and drops below this value if the variables are dependent. The values of $H_{\alpha }^{*}\left(Z\right)$ in Table 1 lead to the same conclusion as those of $S_{i}$, of $x_{2}$ being the most important here, followed by $x_{1}$. The estimator $H_{\alpha }^{*}\left(Z\right)$ neither has the sum property nor does it compute estimates for interaction effects.

In practice, it might be beneficial to compute all the Sobol indices when the input variables are independent and not large in number. However, our prime interest in this study is in general cases with dependent input variables given to us in the form of datasets. In these cases, Sobol indices are difficult to calculate, especially if one wants to take the dependencies into account. By contrast, $H_{\alpha }^{*}\left(Z\right)$ can be computed easily for dependent variables. For dependent variables, the expressions for the Sobol indices become more comprehensive and the interpretation becomes less straightforward, due to the possibility of negative values for the indices.

### 5.2. The El Niño Dataset

In this test case, we considered data of oceanographic and meteorological quantities collected with the Tropical Atmosphere Ocean (TAO) array. This array consists of around 70 buoys in the equatorial Pacific, registering date, latitude, longitude, zonal winds, meridional winds, relative humidity, air temperature and sea surface temperature. The data were obtained from the UCI Machine Learning Repository [35]. The sea and air temperature are known to have a positive correlation.

We explored the dependencies among the winds, humidity, air and sea temperatures. However, before discussing the results, we want to point out a phenomenon frequently occurring in real-world datasets. The measuring equipment registers only a few significant numbers. Therefore, several measurements are registered at the same value, although, in practice, their value was likely to be somewhat different. We correct for this by adding a small noise, such that they end up with slightly different values, before applying the rank transform.

The resulting values for $H_{\alpha }^{*}\left(Z\right)$ are given in Figure 14. The dataset consists of 93,935 data points, hence, we can see in Figure 8 that a value of approximately −0.43 corresponds to independent data.

We see that small dependencies are detected between most variables. The strong dependency between air temperature and sea temperature clearly stands out. It is consistent with prior knowledge [35] and confirms the physical intuition that heat exchange at the air–sea interface results in positive correlation between these temperatures. The dependency between air temperature and relative humidity is less pronounced and physically more complicated, yet consistent with the (negative) correlation between the two that is reported by Pfahl and Niedermann [36] for the equatorial Pacific.

Furthermore, while no linear relationship exists between the zonal and meriodional wind variables, Figure 14 shows weak dependency between them (in agreement with the scatter plot, not shown here). Finally, we note the dependence between zonal wind and sea surface temperature visible in Figure 14. A physical understanding of this dependency involves consideration of the El Niño - Southern Oscillation, which is not discussed in detail here.

## 6. Discussion

In this study, we have combined several methods to come to an efficient approach for quantifying dependencies in data by means of the Rényi mutual information. With this approach, large multivariate datasets can be handled. The Rényi mutual information (MI) is used for the quantification, in which MI is estimated via a minimum spanning tree (MST). The MST can be approximated to speed up the computation. A major advantage of this combination is that it does not require estimation (or a priori knowledge) of probability densities.

Furthermore, we propose a novel estimator for the case where the ranking of the dependencies is required rather than their exact strengths. In this case, the computation of a constant bias term can be omitted.

We tested our approach on the Ishigami function as well as on a real-world test case involving the El Nino dataset. In both cases, we obtained rankings of dependencies consistent with prior knowledge or heuristic understanding. Moreover, in the Ishigami test case, we included an example with independent input variables, making it possible to compare our approach with the analysis based on Sobol indices. The results from our proposed method are consistent with those from the Sobol indices.

Altogether, the approach proposed here is a suitable and useful method to quantify dependencies. It gives consistent results and remains computationally feasible even for large datasets by using the FMST algorithm. In a future study, we aim to extend this method to a sensitivity analysis method by constructing sensitivity indices from the Rényi divergence. An interesting aspect of this extension will be to include interaction effects in these indices.

## Figures and Tables

**Figure 1 entropy-21-00100-f001:**
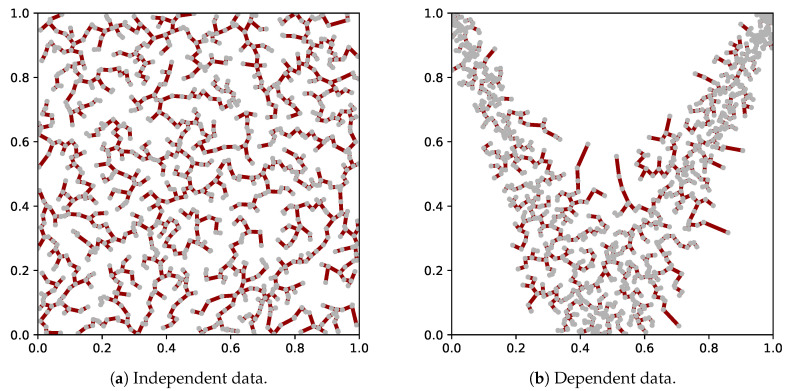
Illustration of the MST for two datasets on the unit square. One dataset is sampled from a bivariate independent distribution (**left**), and the other from a strongly nonlinear dependent distribution (**right**).

**Figure 2 entropy-21-00100-f002:**
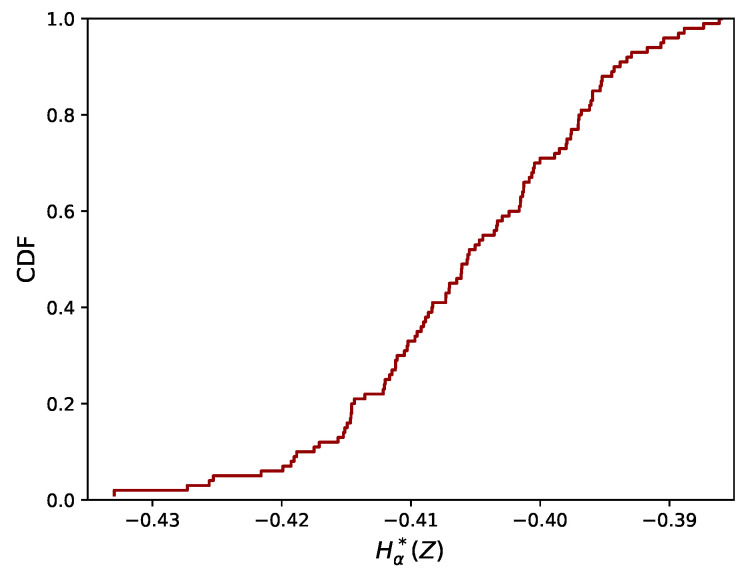
Empirical distribution of $H_{\alpha }^{*}\left(Z\right)$ (Equation (Equation 10)) for the bivariate uniform distribution.

**Figure 3 entropy-21-00100-f003:**
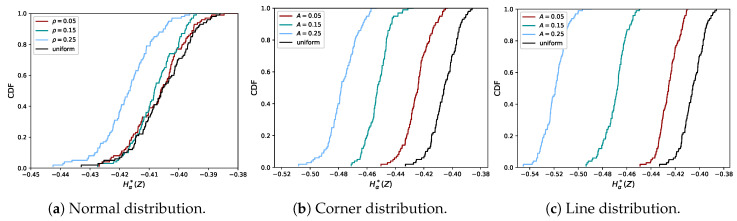
Empirical distributions of $H_{\alpha }^{*}\left(Z\right)$ for data from the normal distribution and the two shape distributions, for small ρ and *A*.

**Figure 4 entropy-21-00100-f004:**
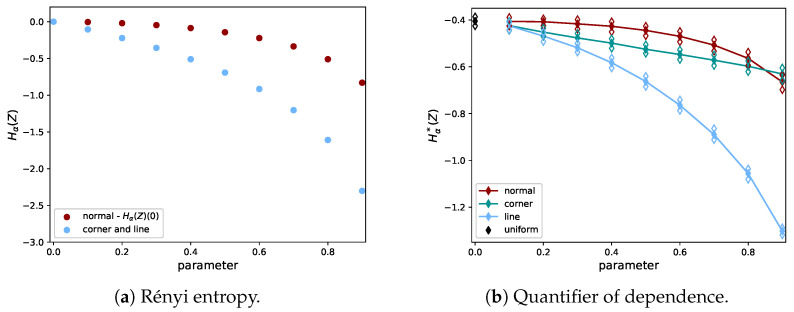
Comparison of the Rényi entropy (**left**) and quantifier of dependence (**right**) for the normal distribution and shape distributions, with varying parameters (ρ and *A*). The Rényi entropy is computed exactly using Equation (Equation 2) without transforming the data. For visualization purposes, the values for the normal distribution have been translated by its value for ρ=0, which is $2\log(2\sqrt{2\pi })$. The quantifier in Equation (Equation 10) was evaluated using data that were sampled from the distribution and then rank-transformed. We show the mean and 95% confidence intervals of $H_{\alpha }^{*}\left(Z\right)$. Note that the numerical values of the entropy and $H_{\alpha }^{*}\left(X\right)$ are not supposed to coincide (cf. Equations (Equation 8) and (Equation 10)).

**Figure 5 entropy-21-00100-f005:**
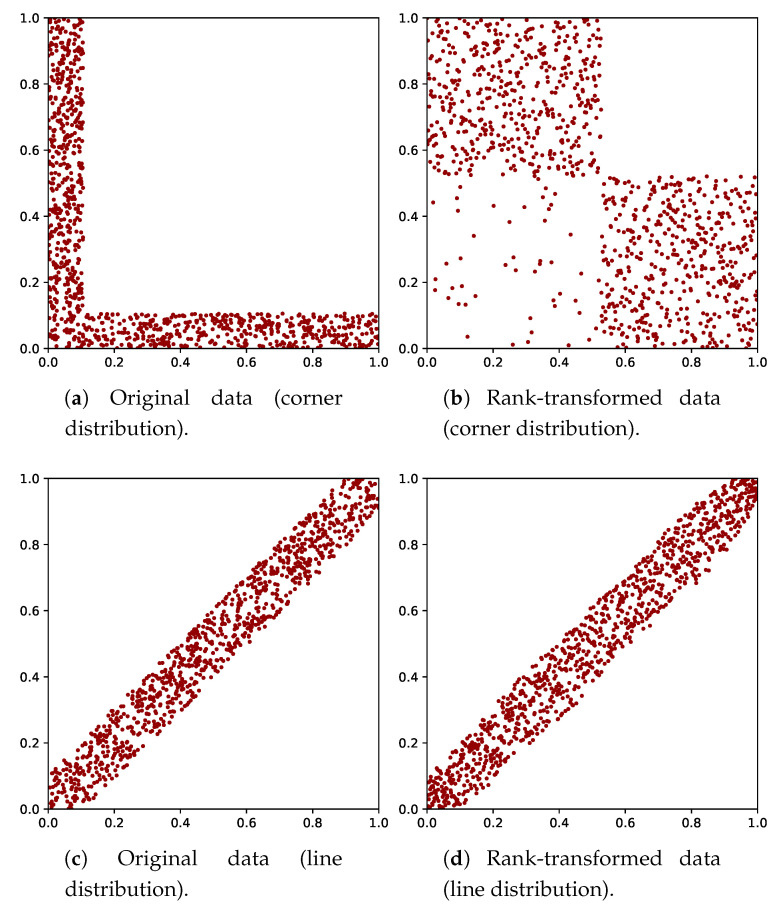
Example of the data and their rank-transform for the shape distribution with A = 0.8.

**Figure 6 entropy-21-00100-f006:**
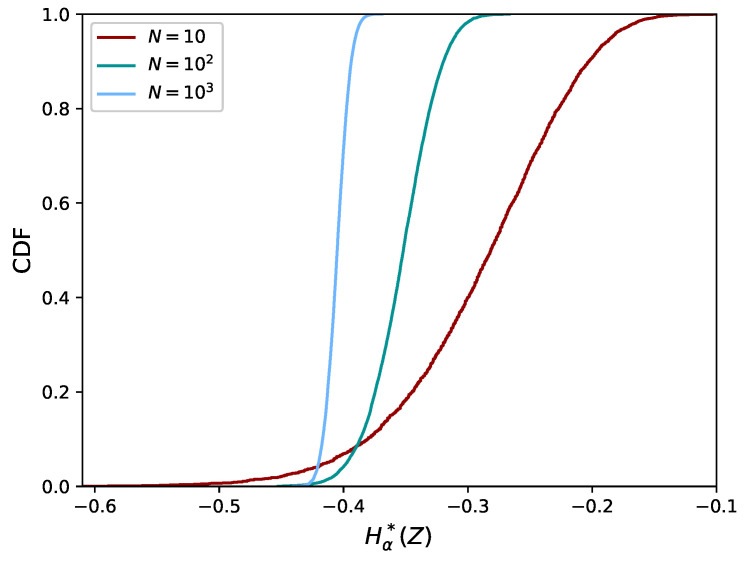
Empirical CDF of $H_{\alpha }^{*}\left(Z\right)$ for the bivariate uniform distribution. The CDF becomes narrower for larger datasets (increasing *N*).

**Figure 7 entropy-21-00100-f007:**
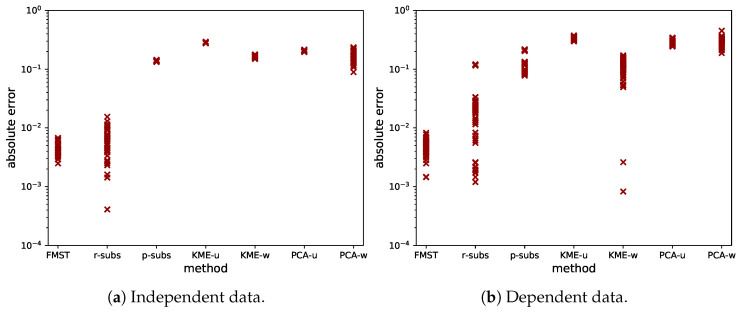
Error estimates for different approximation methods. The figure on the left is for independent data, while the figure on the right is made with dependent data. From left to right, the methods are: FMST, random sampling from subsets-based MST, stratified sampling from subsets-based MST, *K*-means cluster-based MST (unweighted and weighted) and PCA-based cluster-based MST (unweighted and weighted).

**Figure 8 entropy-21-00100-f008:**
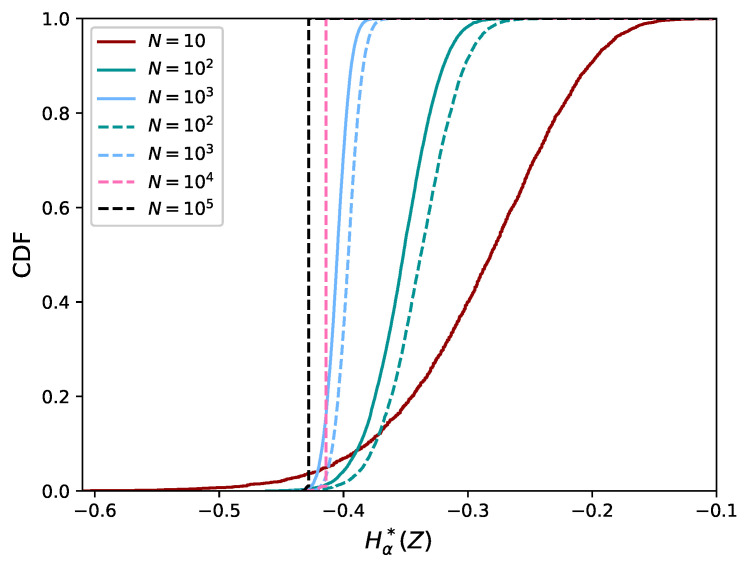
Empirical distribution for the uniform distribution for varying *N*. The solid lines refer to the distributions based on the MST, while the dashed lines refer to the distributions based on the FMST. Results using MST are limited to $N\leq 10^{3}$ because of high computational cost.

**Figure 9 entropy-21-00100-f009:**
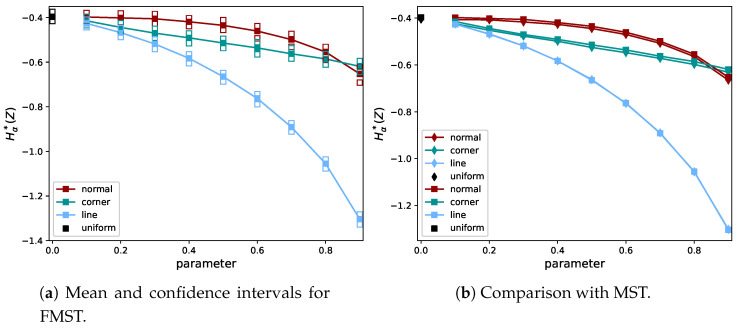
Behavior of the FMST estimator for two different types of data distribution. On the left the mean and empirical 95%-confidence intervals, while on the right the means of the MST (diamonds) and FMST (squares) estimates can be compared (see also Figure 4).

**Figure 10 entropy-21-00100-f010:**
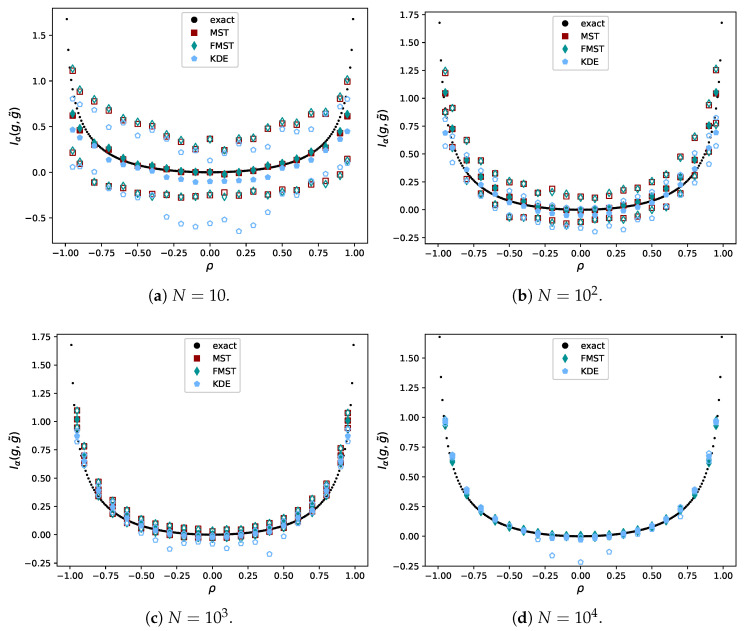
Comparison of the estimated value to the exact value of the divergence.

**Figure 11 entropy-21-00100-f011:**
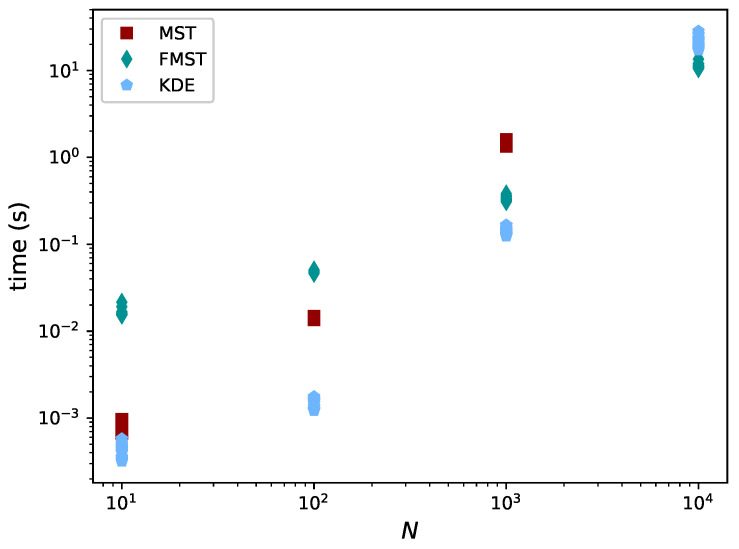
Empirical CDF of $H_{\alpha }^{*}\left(X\right)$ for the bivariate uniform distribution. The CDF becomes narrower for larger datasets (increasing *N*).

**Figure 12 entropy-21-00100-f012:**
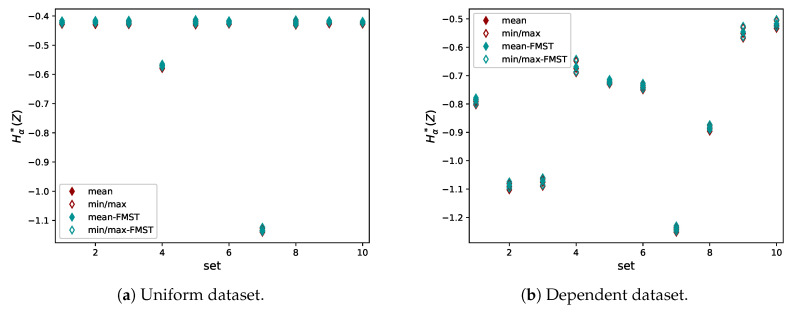
Estimates of $H_{\alpha }^{*}\left(Z\right)$ for two datasets. Note the difference in the range of the *y*-axis.

**Figure 13 entropy-21-00100-f013:**
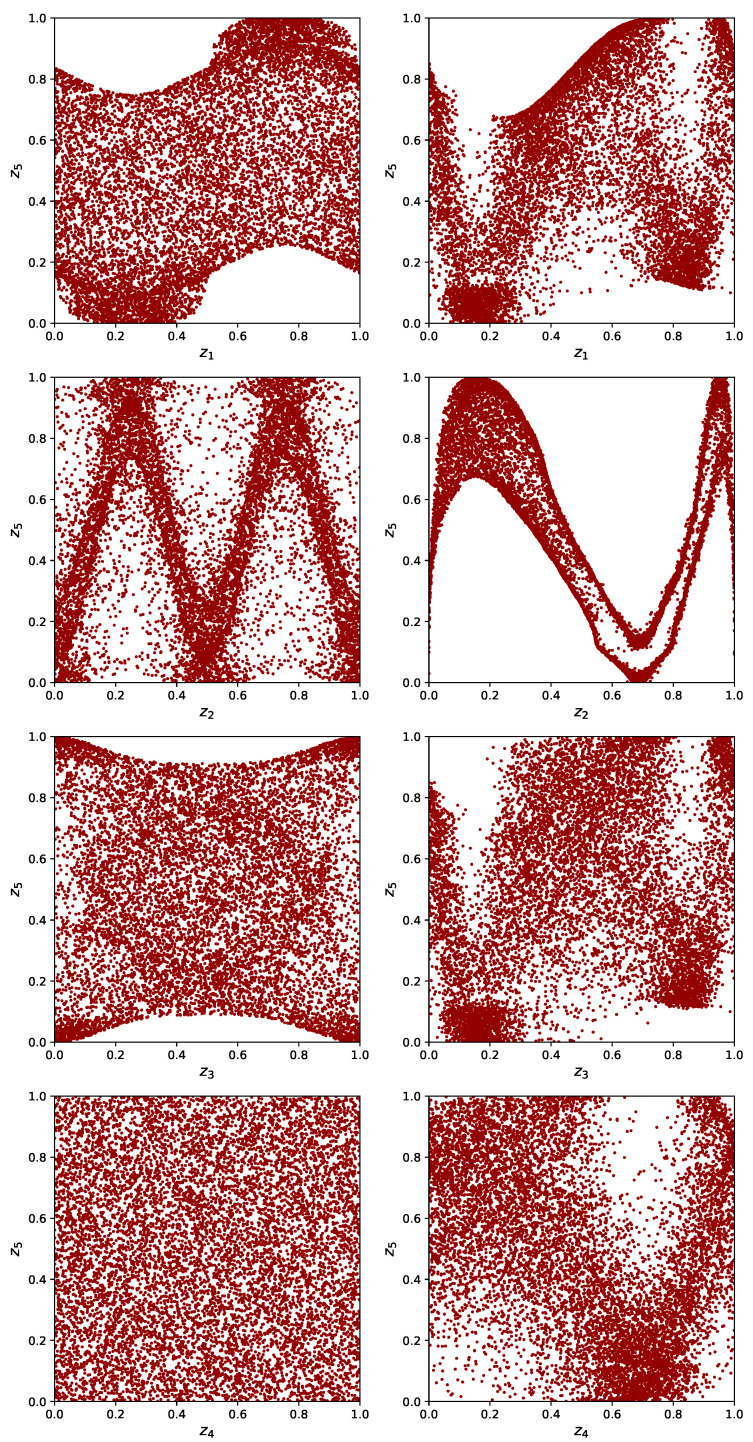
Scatterplots for the Ishigami function with on the left the uniform dataset and on the right the strongly dependent dataset. The dependent input data have a large effect on the output distribution.

**Figure 14 entropy-21-00100-f014:**
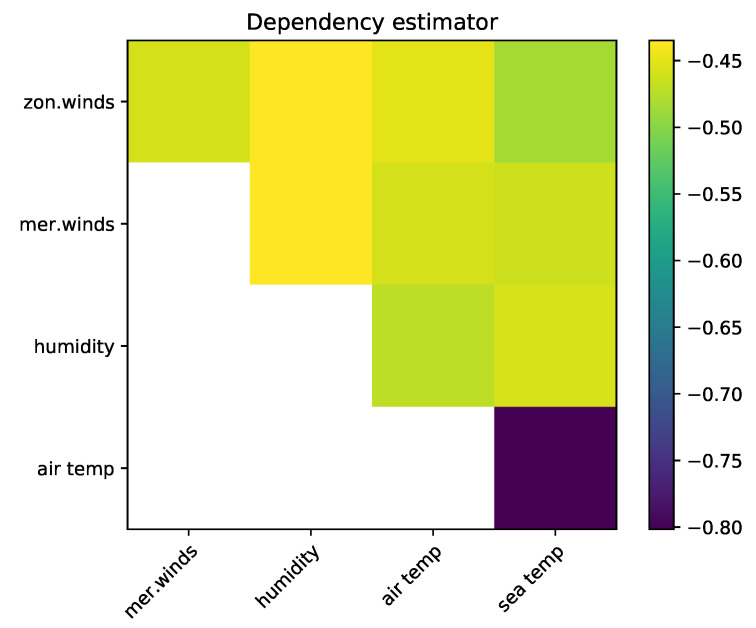
Values of $H_{\alpha }^{*}\left(Z\right)$ for the El Nino test case.

**Table 1 entropy-21-00100-t001:** Comparison of the proposed estimator and Sobol indices for the Ishigami function with independent (uniformly distributed) input variables.

	$S_{i}$	$S_{\mathrm{Ti}}$	$H_{\alpha }^{*}\left(Z\right)$
$x_{1}$	0.314	0.558	−0.570
$x_{2}$	0.442	0.442	−1.133
$x_{3}$	0	0.244	−0.420
$x_{4}$	0	0	−0.421

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
