# Peer review of "Quantifying Data Dependencies with Rényi Mutual Information and Minimum Spanning Trees"

_entropy, 2019, doi:10.3390/e21020100_

Round 1

Reviewer 1 Report

See file here attached

Author Response

The reply is in the PDF file.

Reviewer 2 Report

Please see the attached comments and suggestions.

Author Response

The reply is in the PDF file.

Reviewer 3 Report

Overall it is a well-written paper. Maybe I am wrong, but if it is possible to compare the proposed methods with those based on estimated distributions? In addition proofread could be conducted to improve the presentation of the paper.

Author Response

The reply is in the PDF file.

Round 2

Reviewer 2 Report

Referee Report

Quantifying Data Dependencies with R\'{e}nyi Entropy and Minimum Spanning Trees by Anne Eggels and Daan Crommelin (2nd Revision)

In the revision, the authors have responded to my comments and produced a document that is a significant improvement with respect to the earlier version. In particular, in the revision, they have clarified their contribution and included additional numerical comparisons. 

The revision adequately addresses the issues raised concerning Issues 1-9.  

In my opening, the paper has been improved significantly and can provide a good contribution to the Entropy Journal. Thus, I am recommending to accept the paper.